# One-instrument, objective microsatellite instability analysis using high-resolution melt

**Kamilla Kolding Bendixen**[1], **Sofie Forsberg-Pho**[1], **Giulia Dazio**[2], **Emeli Elisabeth Hansen**[1], **Sarah Kronborg Eriksen**[1], **Samantha Epistolio**[2], **Elisabetta Merlo**[2], **Renzo Boldorini**[3], **Tiziana Venesio**[4], **Alessandra Movilia**[5], **Cecilia Caprera**[6], **Eva Christensen Arnspang**[7], **Michael Børgesen**[1], **Ulf Bech Christensen**[1], **Milo Frattini**[2☯], **Rasmus Koefoed Petersen**[1☯*]

**1** PentaBase A/S, Odense, Denmark, **2** Institute of Pathology, Ente Ospedaliero Cantonale, Locarno, Switzerland, **3** Unit of Pathology, Department of Health Sciences, University of Eastern Piedmont, Novara, Italy, **4** Candiolo Cancer Institute, Fondazione del Piemonte per l'Oncologia, Candiolo, Italy, **5** Hospital of Legnano, SS Biologia Molecolare, UO Anatomia Patologica, Azienda Socio Sanitaria Territoriale Ovest Milanese, Ospedale di Legnano, Legnano, Italy, **6** Laboratory of Molecular Oncology and Predictive Medicine, Pathology Unit, Azienda Ospedaliera Santa Maria di Terni, Terni, Italy, **7** Department of Green Technology, Faculty of Engineering, University of Southern Denmark, Odense, Denmark

☯ These authors contributed equally to this work.
* rkp@pentabase.com

**Data Availability Statement:** All relevant data are within the manuscript and its Supporting Information files.

## Abstract

In recent years, immune checkpoint inhibitors have proved immense clinical progression in the treatment of certain cancers. The efficacy of immune checkpoint inhibitors is correlated with mismatch repair system deficiency and is exceptionally administered based exclusively on this biological mechanism independent of the cancer type. The promising effect of immune checkpoint inhibitors has left an increasing demand for analytical tools evaluating the mismatch repair status. The analysis of microsatellite instability (MSI), reflecting an indirect but objective manner the inactivation of the mismatch repair system, plays several roles in clinical practice and, therefore, its evaluation is of high relevance. Analysis of MSI by PCR followed by fragment analysis on capillary electrophoresis remains the gold standard method for detection of a deficient mismatch repair system and thereby treatment with immune checkpoint inhibitors. Novel technologies have been applied and concepts such as tumor mutation burden have been introduced. However, to date, most of these technologies require high costs or the need of matched non-tumor tissue as internal comparator. In this study, we present a novel, one-instrument, fast, and objective method for the detection of MSI (MicroSight® MSI 1-step HRM Analysis), which does not depend on the use of matched non-tumor tissue. The assay analyzes five well-described mononucleotide microsatellite sequences by real-time PCR followed by high-resolution melt and evaluates microsatellite length variations via PCR product melting profiles. The assay was evaluated using two different patient cohorts and evaluation included several DNA extraction methodologies, two different PCR platforms, and an inter-laboratory ring study. The MicroSight® MSI assay showed a high repeatability regardless of DNA extraction method and PCR platform, and a 100% agreement of the MSI status with PCR fragment analysis methods applied as clinical comparator.

**Funding:** This work was supported by Innovation Fund Denmark (www.innovationsfonden.dk) [grant number 8062-00374B and 9078-00239B] in the form of research materials and salaries (K.K.B., E. E.H., S.K.E., M.B., U.B.C., and R.K.P). Innovation Fund Denmark did not have any additional role in the study design, data collection and analysis, decision to publish, or preparation of the manuscript.

**Competing interests:** I have read the journal's policy and the authors of this manuscript have the following competing interests: K.K.B., E.E.H., S.K. E., M.B., U.B.C., and R.K.P. received salaries from Innovation Fund Denmark. The funding organization did not play a role in the study design, data collection and analysis, decision to publish, or preparation of the manuscript and only provided financial support in the form of authors' salaries and research materials. K.K.B., S.FP., E.E.H., S.K.E., M.B., U.B.C., and R.K.P. were employed at PentaBase A/S during this study. PentaBase A/S is the manufacturer and seller of MicroSight® MSI 1-step HRM Analysis, which is the assay of interest in this study. K.K.B., E.E.H., S.K.E., U.B.C., and R.K. P. are all founders of a patent application covering the novel technology which MicroSight® MSI 1-step HRM Analysis is based on [International Publication number: WO2020229510A1]. The remaining authors declare no conflict of interest. This does not alter our adherence to PLOS ONE policies on sharing data and materials.

## 1. Introduction

Microsatellites are sequences comprised of multiple repeats of the same 1–6 nucleotide sequences. Like other DNA copying errors occurring during replication, errors in microsatellites are normally repaired by the mismatch repair (MMR) system [1]. The MMR system is a complex mainly composed of four proteins working as heterodimers: MutL protein homolog 1 (MLH1)/PMS1 homolog 2 (PMS2) and mutS homolog 2 (MSH2)/mutS homolog 6 (MSH6). Inactivation or reduced efficiency of the MMR system, which can be observed in several human cancer types, may lead to permanent genomic alterations in the length of microsatellites, causing microsatellite instability (MSI) [2, 3]. Microsatellite regions are usually invariant in healthy individuals but polymorphic across populations [4].

MSI occurs in approximately 15% of colorectal cancers (CRC), one fifth of these cases occurring in patients affected by Lynch Syndrome (LS) and the remaining in sporadic cases [5]. Identification of MSI is important in the diagnosis of LS, as a prognostic marker, and as a potential predictive factor of response to chemotherapy and immune checkpoint inhibitors (ICIs). Indeed, several studies have reported that high MSI (MSI-H) CRCs have a better overall prognosis than CRCs characterized by microsatellite stability (MSS) [6–8]. Moreover, multiple studies have reported that stage II MSI-H CRCs do not benefit from 5-fluorouracil (5FU)-based chemotherapy [9] whereas stage II MSS or low MSI (MSI-L) CRC respond well to this type of treatment [2]. More recently, ICIs have demonstrated great efficacy in cancer patients with MSI-H, not only in CRC, leading the Food and Drug Administration (FDA) and the European Medicines Agency (EMA) to approve the treatment of any MSI-H solid tumor with ICIs [10, 11]. Therefore, a precise and if possible objective method for the evaluation of MSI is needed.

The evaluation of MSI is an indirect method to assess the efficiency of the MMR system. The assessment of whether the MMR pathway is deficient in a given patient can be done by measuring the level of expression of the four proteins involved in that pathway, with absence of expression meaning MMR system deficiency. However, the immunohistochemical evaluation of MMR proteins is often challenging [12], due to the site of activity of these proteins (inside the nucleus, leading to the need for strong unmasking of antigens), and several institutions prefer to perform the analysis of MSI on DNA. Many technologies have therefore been developed for MSI analysis, including polymerase chain reaction (PCR) followed by fragment analysis on capillary electrophoresis, herein referred to as PCR fragment analysis, or next generation sequencing (NGS), with the latter developed only recently [13, 14]. DNA based evaluation of MMR system deficiency will not only depend on the method applied, but certainly also the pre-analytical conditions and the resulting concentration, purity, and integrity of sample, as clearly demonstrated in a recent contribution [15].

The Bethesda panel, used with the PCR fragment analysis, has been the gold standard panel for MSI testing in CRC and LS-related cases for over 20 years [16, 17]. The panel consists of two mononucleotide loci (repeats of a single nucleotide), BAT25 and BAT26, and three dinucleotide loci (repeats of two nucleotides), D2S123, D5S346, and D17S250. Dinucleotide loci are highly polymorphic among individuals [17] and, therefore, the analysis of matched non-tumor tissue to tumor tissue is mandatory. On the contrary, mononucleotide loci have a greater sensitivity and specificity [18, 19] and show very low polymorphism within different populations (quasi-monomorphism, less than 1% in Caucasian and approximately 10% in African and Afro American populations) [20]; therefore, the analysis of the matched non-tumor tissue can be avoided. Consequently, the analysis of normal DNA samples in panels that include only mononucleotide repeats (BAT25, BAT26, and new ones more recently discovered such as NR21, NR22, and NR24) can be theoretically avoided and substituted with a universal

DNA reference [21]. In recent years, a panel comprising the mononucleotides BAT25, BAT26, NR21, NR24, and MONO27 has outcompeted the Bethesda panel in terms of performance, and its use in clinical diagnosis is progressively increasing [19, 22, 23].

In this study, we present a novel, one-instrument assay (MicroSight® MSI 1-step HRM Analysis, PentaBase A/S, Odense, Denmark, hereafter referred to as MicroSight® MSI), which investigates five quasi-monomorphic mononucleotide repeats (BAT25, BAT26, NR22, NR24, and MONO27) and returns objective results in less than two hours. The assay utilizes real-time PCR followed by high-resolution melt (HRM) where PCR products are analyzed based on their melting profile, which varies according to GC-content, sequence, length, and heterozygosity. The mononucleotide repeats are analyzed using highly specific probes (Easy-Beacon™ probes) in which a DNA analog platform called Intercalating Nucleic Acid (INA®) is employed [24]. INA®s affect the π-stacking of DNA leading to a stronger binding of the probes to the repeats and thereby an increased specificity and melting temperature, which allows for detection of MSS/MSI. In the present study, MicroSight® MSI data are compared to results obtained using the standard fragment analysis procedure. The assay is also validated for DNA extracted using several extraction methods, as well as on two different PCR platforms, and in an inter-laboratory ring study involving five European institutions.

## 2. Samples and methods

### 2.1. Samples and cohorts

For this study, the DNA extracted from 185 patients diagnosed with CRC was used. The MSI status of the development cohort was characterized at the Institute of Pathology EOC of Locarno (Switzerland). All 185 CRC samples included were consecutive and collected between 2016 and 2017. All patients were anonymized, and the study was approved by the Institutional Ethical Committee of the Institute of Pathology of Locarno (Switzerland). Moreover, all procedures were performed according to the ethical standards of the Helsinki Declaration. The development cohort was analyzed in Locarno, Switzerland (laboratory 1), and the data was utilized to define the settings of MicroSight® MSI. The cohort included 33 MSI-H, 1 MSI-L, and 151 MSS samples. From these patients, 30 cases were selected and subsequently analyzed in four other laboratories located in Italy (Legnano, Novara, Candiolo, and Terni). Furthermore, the assay was validated by analyzing a validation cohort comprising another 127 in-house CRC samples: 30 CRC samples in laboratory 2 (Legnano), 31 CRC samples in laboratory 3 (Novara), 30 CRC samples in laboratory 4 (Candiolo), and 36 CRC samples in laboratory 5 (Terni). The validation cohort consisted of 37 MSI-H, 11 MSI-L, and 79 MSS samples. Samples from both the development and the validation cohorts were previously characterized by the PCR fragment analysis.

An overview of the cohorts is illustrated in Fig 1

### 2.2. Analysis of development cohort by PCR fragment analysis

PCR fragment analysis of the development cohort was conducted in Laboratory 1 using Plenti-Plex™ MSI PentaBase Panel (PentaBase A/S) which includes the following mononucleotide loci (BAT25, BAT26, NR22, NR24, and MONO27). Genomic DNA from both tumor and normal mucosa tissues was extracted from formalin-fixed paraffin-embedded (FFPE) material following the QIAamp DNA FFPE tissue kit protocol (Qiagen, Chatsworth, CA, USA) (herein referred to as QIAamp) and was quantified by Nanodrop 1000 (Witec, Littau, Switzerland). The FFPE sections (thickness: 3–4 μm) contained at least 20% tumor cells.

Fifty nanograms of genomic DNA from each patient were amplified using a multiplex, ready-to-use PCR Mastermix (PlentiPlex™ MSI PentaBase Panel, PentaBase A/S) approach. The panel includes five mononucleotide loci, with the forward primers fluorescently labelled

## A. Development cohort

## B. Validation cohort

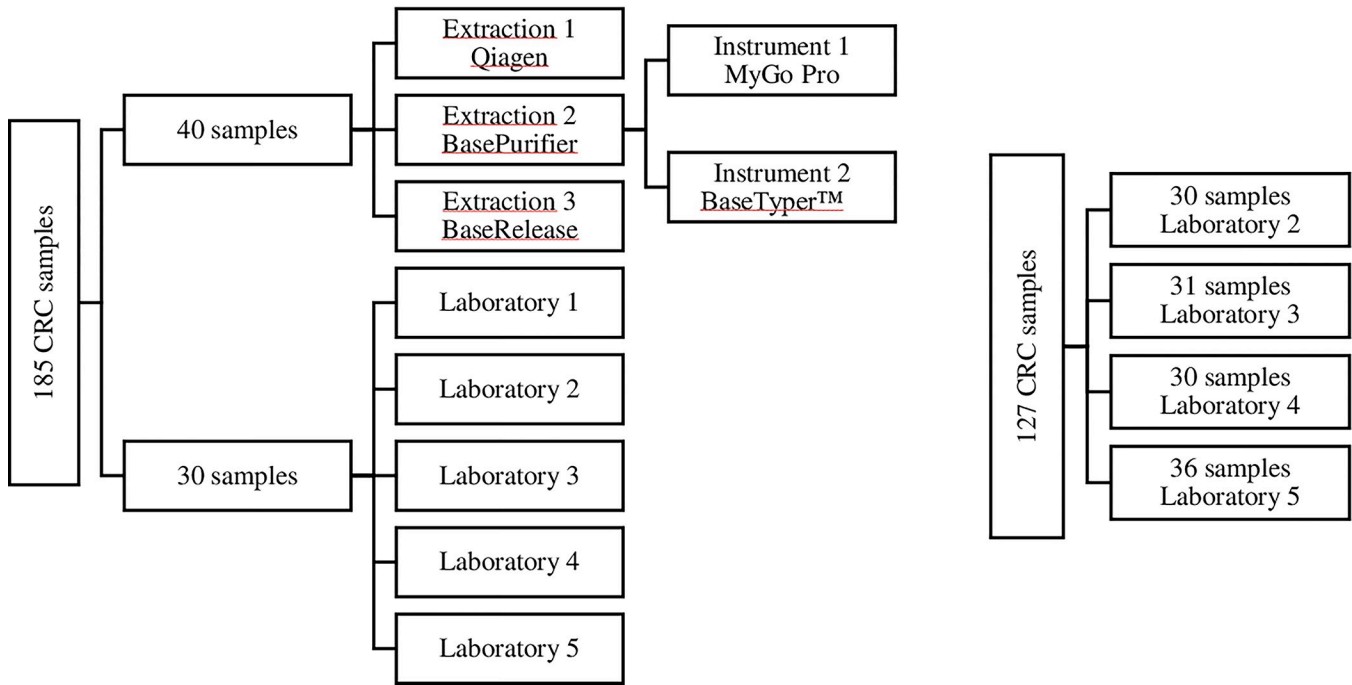

**Fig 1. Overview of cohorts.** A) Overview of the development cohort. A total of 185 colorectal cancer (CRC) samples were analyzed. Forty of these samples were tested on a total of three extraction method and two different real-time PCR instruments. Thirty of the samples were analyzed in a total of five laboratories (1; Locarno, 2; Legnano, 3; Novara, 4; Candiolo, and 5; Terni). B) The validation cohort covered 127 CRC samples distributed in four laboratories.

(S1 Table). The following PCR program was used for amplification: initial denaturation for 5 min at 95˚C, 10 cycles of 3-step amplification with 50 s at 94˚C, 50 s at 55˚C, and 50 s at 72˚C, 25 cycles of 3-step amplification with 30 s at 89˚C, 30 s at 55˚C, and 30 s at 72˚C, 10 min hold at 72˚C, and infinity hold at 10˚C.

The PCR products were diluted 1:10 in water and 1 µL was then mixed with 10 µL of HiDi™ Formamide (Applied Biosystems, Foster City, CA, USA) and 0.25 µL of the Genescan-500LZ DNA ladder (Applied Biosystems). The solution was incubated at 95˚C for 2 min and then at 4˚C for at least 10 min. Finally, the samples were subjected to capillary electrophoresis using a 3130 Genetic Analyzer (Applied Biosystems). Data was analyzed comparing spectrograms between normal and tumor tissue using the Gene Mapper software (Applied Biosystems). Instability is defined by the presence of one or more additional alleles in the tumor tissue. Based on the number of loci displaying instability, the patient can be classified as MSI-H, when more than 30% of the markers are unstable; MSI-L, when at least one locus but less than 30% of all markers are unstable, and MSS, when the patient is stable in all loci [17].

These results were considered as reference and were used to calculate sensitivity and specificity for the MicroSight® MSI assay.

### 2.3. MicroSight® MSI standard set-up

The development cohort was also analyzed using MicroSight® MSI on a MyGo Pro real-time PCR instrument (IT-IS Life Science Ltd., Dublin, Ireland), herein referred to as MyGo Pro.

MicroSight® MSI is a PCR-based method utilizing HRM to obtain the MSI status of tumor samples by analyzing five mononucleotide loci: BAT25, BAT26, NR22, NR24, and MONO27. Five ng of the extracted normal and tumor DNA from each patient from the development cohort were amplified in simplex reactions. Five ng of universal reference purified from a human blood donor were either amplified in the same run as the tumor DNA or afterwards merged into the PCR data file. The following PCR program was used for amplification and endpoint HRM analysis: Initial denaturation for 2 min at 95˚C, 60 cycles of 3-step amplification with 10 s at 95˚C, 10 s at 55˚C, and 10 s at 72˚C, a pre-melt hold for 10 s at 95˚C and HRM from 37–70˚C at 0.05˚C/s. The primer sets created amplicons of 79–107 base pairs in length. For each locus, one EasyBeacon™ fluorescent probe covering the mononucleotide sequence was used to detect the difference in length of normal (or universal reference) and tumor tissue DNA.

## 2.4. MSI classification by MicroSight® MSI

For the paired samples, MicroSight MSI classification reflected the fragment analysis calling; all loci being stable gave the status MSS, one unstable locus was indicative of MSI-L, and two or more unstable loci resulted in microsatellite MSI-H status. For unpaired samples using a universal reference instead of normal DNA, zero to two out of five loci being unstable classified as MSS, and three or more unstable loci gave MSI-H.

When applying a universal reference, it will often be from homozygote DNA, and therefore, a tumor with heterozygote loci will be classified as unstable. Consequently, it is not possible to characterize tumors as MSI-L utilizing a universal reference. However, due to the low frequency of heterozygosity in quasi-monomorphic loci, the possibility to have more than one locus with heterozygosity is negligible and misclassification of MSI-H due to intrinsic heterozygosity is quite null. As a further precaution, tumor samples must be unstable in at least three loci to be characterized as MSI-H [19, 21] when using a universal reference with the MicroSight® MSI assay (more than 40% of the loci measured unstable).

## 2.5. Analysis of development cohort with MicroSight® MSI

The HRM data from the development cohort was used to determine the settings for the analysis of the MSI-status. The stability status of each locus was compared to the results from the PlentiPlex™ MSI analysis. The following analysis parameters were set for the HRM analysis: initial and final normalization area, noise reduction, normalization method (bilinear vs. normal), and temperature shift intensity. A threshold was set for calling the MSI status based on the minimum relative fluorescence unit (RFU) value of the difference plot. MyGo Pro software version 3.5.12+pentabase (PentaBase A/S) was used for the MSI analysis.

A universal reference sample was extracted from whole blood using the Whole Blood Genomic DNA Extraction Kit (Xi'an TianLong Science and Technology, Shaanxi Province, China) on the BasePurifier™ Nucleic Acid Extraction Instrument (PentaBase A/S). The HRM curves of the tumor samples were merged in the software with HRM curves from a universal reference analyzed in a separate run on another MyGo Pro. The settings were optimized to obtain the highest sensitivity and specificity possible.

## 2.6. Comparison of extraction methods

To test the robustness of the MicroSight® MSI assay, DNA extracted by three different methods was evaluated. Forty paired FFPE sections from samples of patients affected by CRC from the development cohort (sections from these patients were previously extracted using the spin-

column-based method, QIAamp) were extracted using two different DNA extraction methodologies: one is a magnetic-bead based method coupled to an automated instrument (BasePurifier™ FFPE, PentaBase A/S) and the other based on a quick DNA release without purification (BaseRelease™, PentaBase A/S). To select the area of interest and moreover to reduce the presence of healthy tissue in the tumor slide, all sections were evaluated and marked by a Pathologist. Genomic DNA from tumor and normal tissues was manually scraped from two slides (3–4 μm thickness each) of FFPE material. Equal amounts of tissues from the patient samples, used for the extraction with the three methods, were secured by macroscopic examination.

The samples were extracted according to the BasePurifier™ FFPE protocol (PentaBase A/S) and to the BaseRelease™ protocol (PentaBase A/S). Samples purified with the BasePurifier™ were diluted five times prior to MicroSight® MSI analysis, and the samples purified with BaseRelease™ were diluted two times. The tumor samples were re-analyzed with a universal reference by merging the run-files with the run-file containing HRM curves from the universal reference.

## 2.7. Comparison of two real-time PCR instruments

The reproducibility of the MicroSight® MSI assay was evaluated on two different PCR platforms. The 40 CRC samples extracted with the BasePurifier™ protocol were re-analyzed with another PCR Instrument, BaseTyper™ 48.4 Quiet HRM Real-Time PCR System (PentaBase A/S), herein referred to as BaseTyper™. The HRM analysis settings were adapted to fit the BaseTyper™ instrument, but the PCR program and analysis were essentially the same. BaseTyper™ software version V1 (PentaBase A/S) was used for automated analysis of the results. Universal reference DNA was included in each PCR run to compare paired samples with the universal reference. The samples were classified according to the minimum RFU value of the difference plot. However, in the BaseTyper™ software the maximum amplitude of the difference plot (either positive or negative) is displayed. This value was used for creating boxplots.

## 2.8. Inter-laboratory variation

Thirty samples from the development cohort that was previously analyzed in laboratory 1 were sent to the four validation laboratories (laboratory 2–5). The samples were shipped from laboratory 1 as purified DNA in a concentration of 1–2 ng/μL. This part of the study only evaluated the inter-laboratory reproducibility of the MicroSight® MSI assay and not the full workflow with DNA extraction. The validation laboratories were trained and analyzed the samples independently using a MyGo Pro.

## 2.9. Analysis of validation cohort

The validation cohort had previously been analyzed by various MSI panels all utilizing the PCR fragment analysis. The samples were extracted using different extraction procedures, depending on the laboratory. Laboratory 2 used FFPE sections containing at least 50% tumor cell for the FPPE extraction using Maxwell® (Promega Corporation, Madison, Wisconsin, USA), and the concentration of DNA was determined using a nanodrop method. The Bethesda panel was used for the MSI analysis. Laboratory 3 used FFPE tumor samples with at least 50% tumor cells and cut 4–5 sections of 5 μm prior to extraction. The samples were extracted using the same procedure as in laboratory 2. They used a panel comprising only mononucleotides (BAT25, BAT26, NR21, NR24, and NR27) for the PCR fragment analysis. Laboratory 4 used QIAamp for extraction of DNA from FFPE tumor samples (not further specified) and OncoMate™ MSI Dx Analysis System (Promega Corporation, Madison, Wisconsin, USA), which covers the five mononucleotide loci BAT25, BAT26, NR21, NR22, and

MONO27, for the PCR fragment analysis. Laboratory 5 used automated nucleic acid extractor (not further specified) and used the panel Titano MSI (Diatech Pharmacogenetics srl, Jesi, Italy) for the PCR fragment analysis covering the Bethesda panel and BAT40, D18S58, NR21, NR24, and TGFβRII. The results from the different PCR fragment analyzes were compared to the results from MicroSight® MSI.

From 1 to 10 ng/μL of purified DNA was used for analysis of the samples with the Micro-Sight® MSI assay using the MyGo Pro.

## 2.10. Data analysis

All data analysis and statistics were made in R studio version 1.3.1093 (R version 4.0.3).

The threshold for calling a locus stable or unstable was individually set for each marker to retain the highest accuracy (sum of sensitivity and specificity) in the development cohort.

The diagnostic sensitivity and specificity were calculated for each locus for MicroSight® MSI. The results from the PCR fragment analysis were used as reference and their sensitivity and specificity were assumed to be 100%. The Wilson score was used to calculate 95% confidence intervals. The sensitivity and specificity could not be called for each locus for the validation cohort, as the reference laboratories used different loci. The agreement in terms of MSI-classification was calculated.

$$Diagnostic\ sensitivity\ [\%] = \frac{True\ Positive}{True\ Positive + False\ Negative} \cdot 100$$

$$Diagnostic\ specificity\ [\%] = \frac{True\ Negative}{True\ Negative + False\ Positive} \cdot 100$$

Unpaired t-test and one-way ANOVA were used to compare the difference in RFU values between various factors expected to influence the RFU difference. This was only applied for the unstable loci, as the RFU value of stable loci are close to zero. The Pearson correlation was calculated to investigate the RFU difference of the same samples analyzed in five different laboratories. Laboratory 1 was set as independent variable. Only unstable loci were included in the analysis. The ANOVA, t-test, and Pearson correlation were calculated on the assumption of normal distributed data.

We made 3x3 contingency tables and computed unweighted Cohen's Kappa to calculate the agreement between the MSI analysis methods both in the development and validation cohort.

## 3. Results

### 3.1. Settings for MicroSight® MSI analysis using paired samples or universal reference

The tumor samples contain a mix of normal cells and tumor cells. If the tumor is characterized by MSI, it will contain both the original length of the microsatellite due to the normal cells and microsatellites with alteration in length from the MSI tumor cells. Mixed length of the micro-satellites affects the shape of the melting curve, and thus it is possible to differentiate between MSS and MSI tumors by comparing the shape of the melting curve of the tumor tissue to the normal tissue [25].

The final settings for the analysis of paired samples and universal reference can be seen in S2 and S3 Tables. The HRM curves were normalized to obtain the same fluorescent intensity of the curves in the initial and final normalization regions. Natural variation in length of

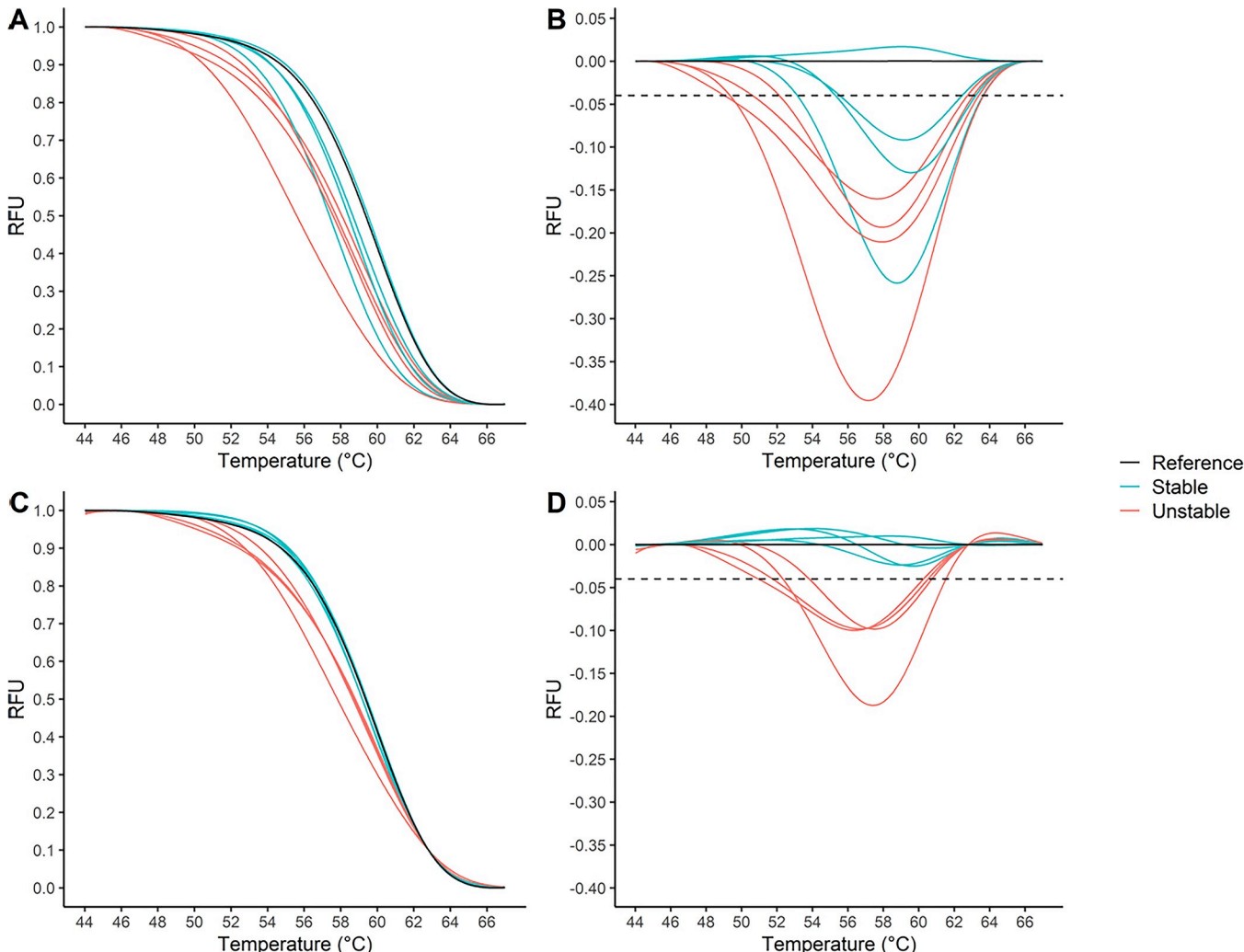

**Fig 2. Principle of MicroSight® MSI analysis using universal reference.** Four stable and four unstable tumor samples were analyzed using a universal reference for NR24 locus. Applying a temperature shift makes it possible to differentiate unstable from stable samples naturally differing in lengths. **A)** Normalized HRM curves without temperature shift. **B)** Difference plot without temperature shift. Threshold was set at -0.04 RFU. **C)** Normalized HRM curves with temperature shift at 0.1 RFU. **D)** Difference plot with temperature shift at 0.1 RFU. Threshold was set at -0.04 RFU. MSI: microsatellite instability; HRM: high resolution melt; RFU: relative fluorescence unit.

microsatellites among stable patients caused a problem for using a universal reference, as exemplified with the NR24 locus (Fig 2A and 2B). Several stable patients were crossing the threshold and were incorrectly called unstable. To compensate for individual differences in melting temperature, the curves were shifted in the x-axis at a set RFU to align the intercept between the intensity threshold and the HRM curves (temperature shift).

A difference plot was created by setting a reference curve as baseline curve, and the baseline curve was subtracted from the tumor curve. Thereby the difference in RFU was plotted as a function of the temperature. If the difference curve crossed a threshold set in the PCR software, the tumor was classified as MSI in the given locus. If the difference did not cross the threshold, the locus was MSS. By applying the temperature shift at a given RFU, it was possible to neutralize these temperature differences between stable patients, and thereby correctly characterize the patients using a universal reference (Fig 2C and 2D).

**Table 1. MicroSight® MSI agreement to the PCR fragment analysis.**

| Type | Extraction method | PCR Instrument | Laboratory | MSS (agreement%) | MSI-L (agreement %) | MSI-H (agreement %) | Cohen's Kappa [95% CI] |
|---|---|---|---|---|---|---|---|
| Paired | QIAamp | MyGo Pro | 1 | 151/151 (100) | 1/1 (100) | 33/33 (100) | 1.000 |
| Paired | BasePurifier | MyGo Pro | 1 | 13/13 (100) | 0/0 (100) | 27/27 (100) | 1.000 |
| Paired | BaseRelease | MyGo Pro | 1 | 13/13 (100) | 0/0 (100) | 27/27 (100) | 1.000 |
| Paired | BasePurifier | BaseTyper | 1 | 13/13 (100) | 0/0 (100) | 27/27 (100) | 1.000 |
| Paired | QIAamp | MyGo Pro | 2 | 25/25 (100) | 0/0 (100) | 5/5 (100) | 1.000 |
| Paired | QIAamp | MyGo Pro | 3 | 25/25 (100) | 0/0 (100) | 5/5 (100) | 1.000 |
| Paired | QIAamp | MyGo Pro | 4 | 25/25 (100) | 0/0 (100) | 5/5 (100) | 1.000 |
| Paired | QIAamp | MyGo Pro | 5 | 25/25 (100) | 0/0 (100) | 5/5 (100) | 1.000 |
| Universal | QIAamp | MyGo Pro | 1 | 151/151 (100) | - | 33/33 (100) | 1.000 |
| Universal | BasePurifier | MyGo Pro | 1 | 13/13 (100) | - | 27/27 (100) | 1.000 |
| Universal | BaseRelease | MyGo Pro | 1 | 12/13 (92.3) | - | 27/27 (100) | 0.9419 [0.8295; 1.000] |
| Universal | BasePurifier | BaseTyper | 1 | 13/13 (100) | - | 27/27 (100) | 1.000 |
| Universal | QIAamp | MyGo Pro | 2 | 25/25 (100) | - | 5/5 (100) | 1.000 |
| Universal | QIAamp | MyGo Pro | 3 | 25/25 (100) | - | 5/5 (100) | 1.000 |
| Universal | QIAamp | MyGo Pro | 4 | 25/25 (100) | - | 5/5 (100) | 1.000 |
| Universal | QIAamp | MyGo Pro | 5 | 25/25 (100) | - | 5/5 (100) | 1.000 |

## 3.2. Analysis of development cohort

The specificity for all loci was 100% using paired samples, and the sensitivity was 100% for all loci except BAT25, which had a sensitivity of 96.97% (95% CI [84.68; 99.46]). The confidence intervals for each locus are shown in S4 Table. However, all samples gave the same MSI-classification as the PCR fragment analysis classifying 151 MSS, 1 MSI-L, and 33 MSI-H (Table 1). For the universal reference, it was not possible to classify as MSI-L, and therefore the one MSI-L sample was categorized as MSS. The specificity varied between 91.2% and 100% and the sensitivity was 100% for all loci except BAT26, having a sensitivity of 94.1% (95% CI [80.91; 98.37]) (S4 Table). The groups of unstable and stable cases (classification according to the PCR fragment analysis results) are illustrated in Fig 3. A gap is seen between the two groups which makes it possible to discriminate between stable and unstable samples. For the samples analyzed with universal reference, several outliers were observed for the stable cases.

## 3.3. Comparison of extraction methods

All MSI-H samples (27/27) were correctly classified using all three extraction methods with paired samples as well as with universal reference and 13/13 MSS samples were correctly classified with paired samples. However, using the universal reference, 13/13 MSS samples were correctly classified using the QIAamp and the BasePurifier™ method, but one MSS sample was classified as MSI-H using the universal reference with BaseRelease™ (Table 1). Low variance in the RFU difference between the extraction methods were observed, this is illustrated for BAT25 in Fig 4. A one-way ANOVA analysis, comparing the RFU difference for the unstable cases for each locus across the three extraction platforms, revealed no significant difference for any of the loci for paired samples (p-values: BAT25 = 0.936, BAT26 = 0.914, NR22 = 0.094, NR24 = 0.629, MONO27 = 0.124) or for unpaired samples (p-values: BAT25 = 0.066, BAT26 = 0.402, NR22 = 0.691, NR24 = 0.245, MONO27 = 0.715).

All three extraction methods had a specificity of 100% (95% CI [77.19; 100.00]) on all loci using paired samples, and the sensitivity was 100% (95% CI [87.54; 100.00]) for all loci except BAT25 on BaseRelease™ which had a sensitivity of 96.3% (CI 95% [81.71; 99.34]). Using the

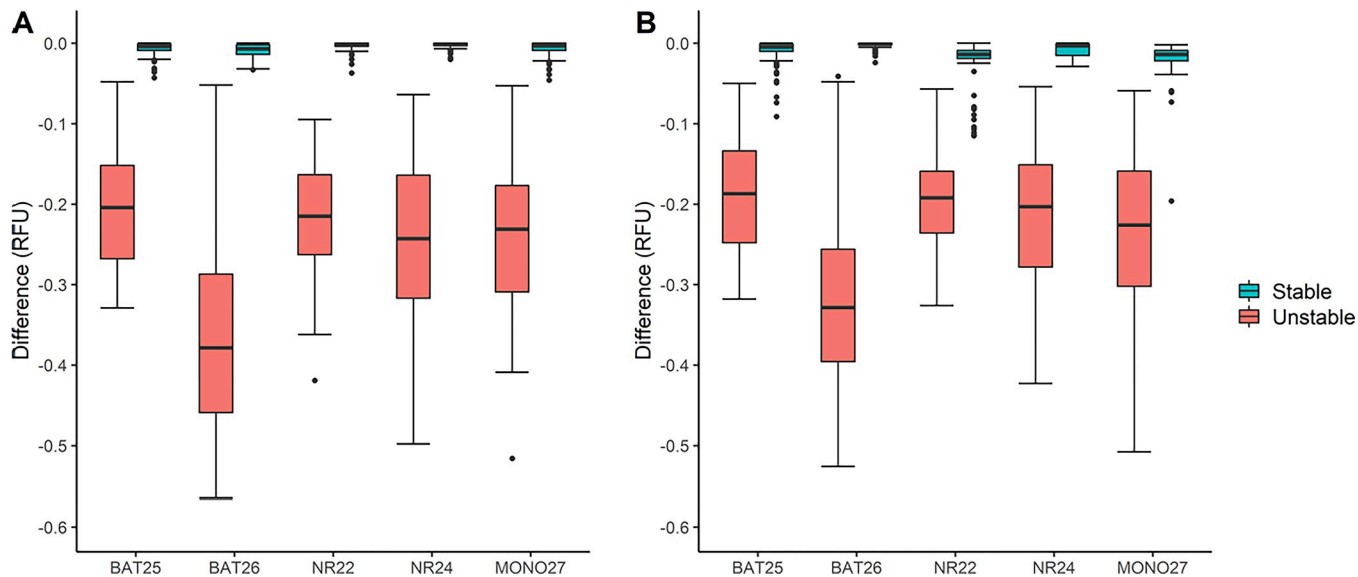

**Fig 3. Boxplot over stable and unstable cases for the development cohort (n = 185).** The difference in RFU for the tumor sample compared to either universal or paired normal samples are plotted for each of the five microsatellite loci. Stable cases represented by light blue boxes and unstable cases by light red boxes as indicated by the figure legends. **A)** Difference using paired normal and tumor samples. **B)** Difference using tumor samples and a universal reference. RFU: relative fluorescence unit.

universal reference, the sensitivity was 100% (95% CI [87.54; 100.00]) for all loci and the specificity varied between 53.9% (CI 95% [29.14; 76.79]) and 100% (95% CI [77.19; 100.00]), with the lowest specificity for BAT25 using BaseRelease™ as extraction method (S5 Table).

### 3.4. Comparison of two real-time PCR instruments

All 40 samples analyzed on the BaseTyper™ were correctly classified according to the PCR fragment analysis classification (Table 1). The settings for the BaseTyper™ can be found in S3 Table. Identical classifications of the samples were obtained on the MyGo Pro and BaseTyper™. The BaseTyper™ software gives the value of the maximum amplitude of the difference plot (either positive or negative) and not only the maximum negative amplitude as in the MyGo Pro software. Therefore, the stable groups covered both negative and positive values using the BaseTyper™ software. The sensitivity and specificity of the loci were the same (100%) for paired samples using both instruments. However, higher specificity was observed using universal reference on the BaseTyper™ (average specificity of 96.9%) compared to the MyGo Pro (average specificity of 90.8%) (S5 Table).

### 3.5. Inter-laboratory variation

All MSS patients (n = 25) and all MSI-H cases (n = 5) were correctly classified using both paired samples and universal reference in all five laboratories (Table 1). All laboratories had 100% agreement for all loci in the analysis of paired samples. Laboratories 1, 3, and 4 had 100% agreement in the analysis of all loci using universal reference. Laboratory 2 found one MSS sample to be unstable in BAT25 and one MSS sample to be unstable in NR24 using the universal reference, which were not found by the other laboratories. Laboratory 5 found one MSS sample unstable in MONO27 using the universal reference (S6 Table). Low variation was observed between the laboratories, which is illustrated using the difference for the unstable

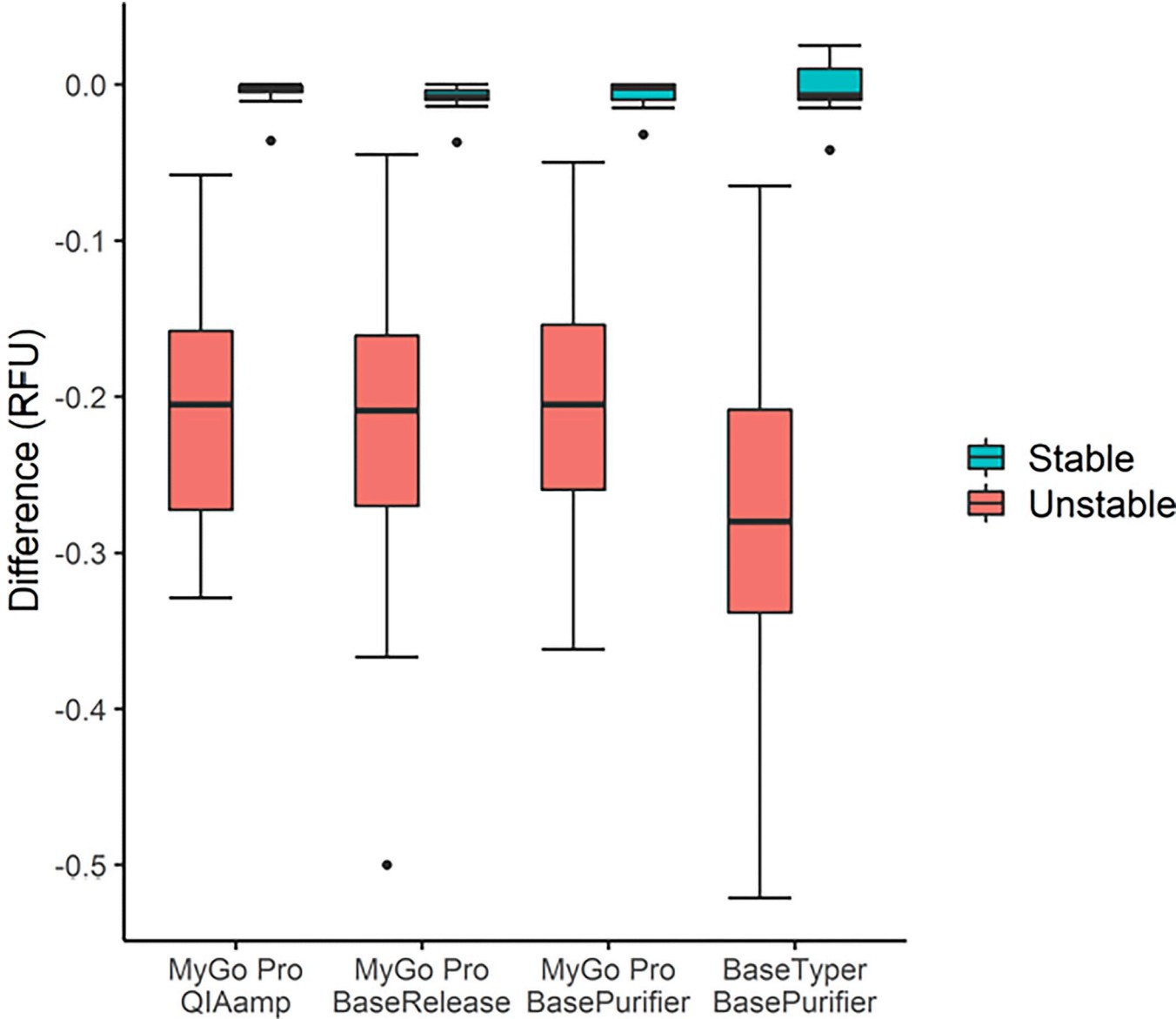

**Fig 4. Results from analysis of paired samples on BAT25 using three extraction methods and two real-time PCR instruments.** Forty CRC samples were extracted using the QIAamp method, BasePurifier™, and BaseRelease™ and analyzed using the MyGo Pro. The samples extracted with BasePurifier™ were re-analyzed using a BaseTyper™ real-time PCR instrument.

cases using paired samples (n = 5) in Fig 5. The Pearson correlation was 0.9779 (95% CI [0.9672; 0.9851], p-value<0.001) for paired samples and 0.9782 (95% CI [0.9678; 0.9853], p-value<0.001) for universal reference.

## 3.6. Validation of MicroSight® MSI

In all four laboratories, there was 100% agreement in the MSI-H samples for both universal reference and paired analysis. An overall 3x3 contingency table for paired samples comparing the MSI-calling of the validation laboratories standard methods and MicroSight® MSI can be found in Table 2. Cohen's Kappa was 0.8053 (95% CI [0.7057; 0.9049] and 0.8019 (95% CI

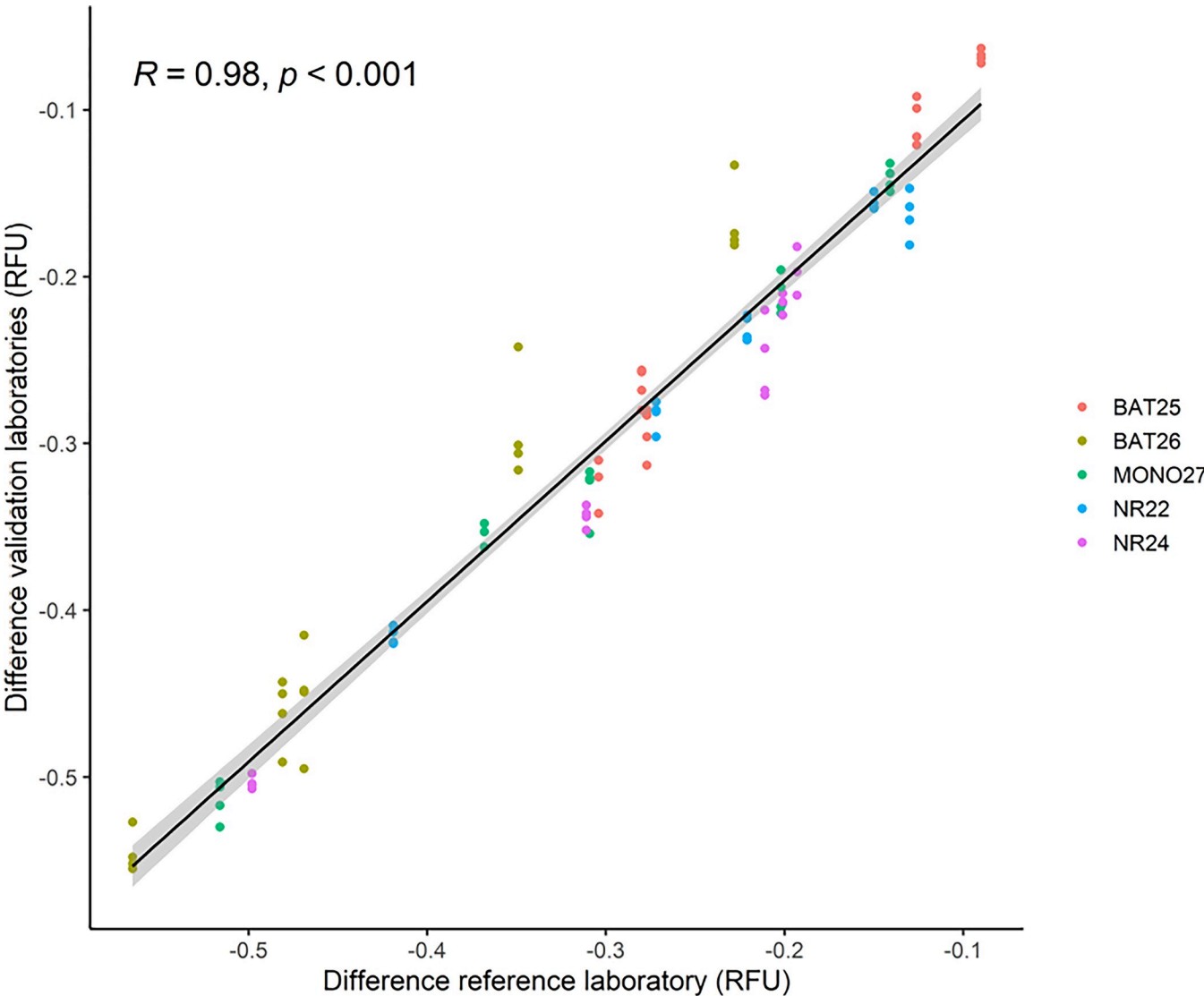

**Fig 5. Correlation plot of unstable cases (n = 5) using paired samples.** The x-coordinates for the points are the difference of RFU obtained in the reference laboratory (laboratory 1) and the y-coordinate is the difference in RFU obtained in the validation laboratories (laboratory 2–5). A linear fit is applied to the points and the Pearson correlation coefficient (R), and p-values are shown. RFU: relative fluorescence unit.

[0.7016; 0.9022]) for paired and universal reference, respectively. 3x3 contingency tables for each individual laboratory can be found in S7–S11 Tables. The agreement in MSI-L was zero except for Laboratory 4 having 20% MSI-L agreement (Table 3) using paired samples. The deviating cases are shown in S12 Table. In Laboratory 2 and 5, five of the six deviating cases came from paired samples called MSI-L by instability in dinucleotide loci in the fragment analysis, and MSS using MicroSight® MSI (no dinucleotide loci). In Laboratory 2, one sample was called MSS by the PCR fragment analysis and MSI-H using MicroSight® MSI both using paired samples and universal reference. In Laboratory 3, one paired sample was called MSI-L due to instability in NR21 in the PCR fragment analysis and was called MSS using the Micro-Sight® MSI assay, which does not include NR21. In Laboratory 4, three paired samples were called MSI-L due to instability in BAT25 using the PCR fragment analysis and was called MSS on MicroSight® MSI, which also includes the BAT25 locus. One paired sample was called

**Table 2. A 3x3 table covering all four validation laboratories (original MSI status) compared to MicroSight® MSI using paired samples.**

| | | MicroSight® MSI | | |
|---|---|---|---|---|
| | | **MSS** | **MSI-L** | **MSI-H** |
| Original status | MSS | 77 | 1 | 1 |
| | MSI-L | 10 | 1 | 0 |
| | MSI-H | 0 | 0 | 37 |

Cohen's Kappa = 0.8053 (95% CI [0.7057; 0.9049])

MSI-L due to instability in NR22 using MicroSight® MSI and was called MSS on the PCR fragment analysis (not covering NR22).

## 4. Discussion

In recent years, the determination of MSI status has acquired more and more importance due to its relevant role as a prognostic and predictive marker, especially from the introduction of ICIs in the armamentarium against cancer and after FDA approval for these therapies in all cancer types [6–8]. At present, the gold standard for assessing the MSI status is based on the PCR fragment analysis [17]. This method is faster and significantly less expensive than the evaluation through a next-generation sequencing approach. However, when it includes both mononucleotide and dinucleotide loci, the availability of matched non-tumor tissue as internal calibrator is needed. To bypass this last requirement, there seem to be other better performing PCR based panels which include the analysis of just mononucleotide loci [21]. In this study we developed and evaluated a new assay for MSI detection, the MicroSight® MSI assay, consisting of five quasi-monomorphic mononucleotide loci (BAT25, BAT26, NR22, NR24, and MONO27). Whereas the clinical benefits of BAT25, BAT26, NR24 and MONO27 loci have been extensively explored, inclusion of NR22 was chosen based on high sensitivity reported [22], the monomorphic nature of the loci [21] and excellent technical performance. MicroSight® MSI differs from the conventional methods as it is a fast, closed-tube assay based on post-PCR melting profile of patient samples utilizing highly specific probes containing INA®s [24].

The efficacy of the new assay was first investigated on a development cohort of 185 CRC patients in the reference laboratory (Laboratory of Molecular Pathology, Institute of Pathology EOC, Locarno, Switzerland). All patient classifications were identical to the results obtained

**Table 3. Agreement of MSI status in validation cohort from four laboratories.**

| | MSS (Agreement %) | MSI-L (Agreement %) | MSI-H (Agreement %) | Cohen's Kappa [95% CI] |
|---|---|---|---|---|
| **Paired** | | | | |
| Laboratory 2 | 21/22 (95.5) | 0/2 (0) | 6/6 (100) | 0.7443 [0.4822; 1.000] |
| Laboratory 3 | 20/20 (100) | 0/1 (0) | 10/10 (100) | 0.9297 [0.7974; 1.000] |
| Laboratory 4 | 13/14 (92.8) | 1/5 (20) | 11/11 (100) | 0.7175 [0.5081; 0.9270] |
| Laboratory 5 | 23/23 (100) | 0/3 (0) | 10/10 (100) | 0.8194 [0.6350; 1.000] |
| **Universal** | | | | |
| Laboratory 2 | 21/22 (95.5) | 0/2 (0) | 6/6 (100) | 0.7443 [0.4822; 1.000] |
| Laboratory 3 | 20/20 (100) | 0/1 (0) | 10/10 (100) | 0.9297 [0.7974; 1.000] |
| Laboratory 4 | 14/14 (100) | 0/5 (0) | 11/11 (100) | 0.7076 [0.5036; 0.9116] |
| Laboratory 5 | 23/23 (100) | 0/3 (0) | 10/10 (100) | 0.8194 [0.6350; 1.000] |

with the PCR fragment analysis. The method was further evaluated in an inter-laboratory ring study with 30 samples from the development cohort analyzed at four different laboratories. All sites reached the same classifications as the reference laboratory. MicroSight® MSI was then validated on at least 30 of the four laboratories' individual samples classified using distinct fragment analysis panels, showing a substantial superimposition between MicroSight® MSI and the other panels. The correlation of MSI-H and MSS status obtained in laboratories 3 and 4, which used panels only comprising mononucleotide loci, was not better than the one obtained in laboratories 2 and 5, which used assays also including dinucleotides. This indicates no substantial difference between the use of methods including only mononucleotide loci in comparison to the use of assays also including dinucleotides. Concerning the MSI-L samples, the agreement was low. This is essentially due to the fact that low instability typically occurs in dinucleotide loci [19], that are evaluated in panels based on the PCR fragment analysis but not in the MicroSight® MSI assay. In addition, MSI-L cases are considered as MSS from the clinical point of view, because the instability observed in just one locus might be due to a random error rather than to absence of activity of the MMR system. As a consequence, the misclassification of MSI-L in MSS does not have any clinical impact and can paradoxically help clinicians who very often have difficulties in the correct clinical interpretation of MSI-L cases [19]. In Laboratory 3, which used a panel with just mononucleotide loci, a tumor was characterized as MSI-L due to instability found in the locus NR21. This sample was classified as MSS by MicroSight® MSI because this assay does not include this locus. Therefore, the only real discrepant case was found by Laboratory 2 which used a different panel but still included the BAT26 locus; this locus appeared to be stable with the own panel of Laboratory 2 and unstable with MicroSight® MSI. A possible explanation can be a different sensitivity of the two assays, being the sensitivity of MicroSight® MSI higher than the one of their own panels.

It has been demonstrated that pre-analytical processes may provide an inaccurate MSI evaluation [15]. Therefore, we also addressed this issue in our study. At first, we assessed the efficacy of the new assay on different DNA extraction methods because it has been reported that those processes can influence the melting temperature of DNA and therefore lead to a possible variation of the final result. Indeed, different studies showing varying results have been carried out to test the robustness of HRM [25, 26]. However, the peculiarity of MicroSight® MSI HRM assay is the focus on the shape of the melting curve and not on the melting temperature itself. This feature resulted in great stability across several extraction platforms, also using a universal reference extracted from whole blood, which led to a complete superimposition of the MSI calling generated from paired DNA extracted with different methods.

Lastly, the new MicroSight® MSI assay was also tested on different qPCR instruments. The BaseTyper™ showed a higher specificity compared to the MyGo Pro utilizing the universal reference. This difference could be explained by the inclusion of a PCR strip containing the universal reference DNA in each PCR run performed on the BaseTyper™. For the MyGo Pro, the data from the universal reference was obtained by merging the data across several PCR instruments and different batches of MicroSight® MSI, and therefore, it was unexpected that the overall results from the universal reference complied with the results from paired samples.

It is crucial for the patient to get the same MSI classification regardless of the laboratory who performs the analysis and the assay used, as the MSI status can determine the choice of treatment, the prognosis, and the possible presence of a hereditary component of the tumor. MicroSight® MSI showed 100% agreement in MSI calling in five independent laboratories and can therefore be considered a robust analysis with low to none inter-laboratory variation. The high agreement is a consequence of the high sensitivity and reproducibility of the assay, and it is related to the fact that MicroSight® MSI characterizes the MSI status of the tumor sample based automatically on a pre-defined threshold.

A fast and unbiased analysis could potentially expand the use of MSI-testing. Furthermore, if a patient's MSI status can be determined using only tumor tissue it will increase the analysis throughput, simplify the overall workflow of MSI determination, and halve the costs for DNA extraction and tissue selection. A standardized method is therefore needed, which will determine the same MSI-status without any subjective analysis with the purpose to administer the right treatment without bias. Utilizing well-known MSI-loci ensures that the method can be directly compared to gold standard methods, and shorter turn-around time can lead to faster results leading to an early treatment of the patient with the appropriate therapy.

## 5. Conclusion

We have developed and validated a novel assay, MicroSight® MSI 1-step HRM Analysis, for the detection of MSI in cancer patients. The assay showed high agreement with the PCR fragment analysis and demonstrated high sensitivity and specificity using both paired samples and universal reference in all laboratories using different extraction methods. Moreover, no discrepancy in the MSI classification was found by comparing two different PCR instruments. To conclude, our data demonstrate that MicroSight® MSI has very similar performance to the fragment analysis, gold standard method, but is superior in ease-of-use, instrument cost and turn-around-time. MicroSight® MSI obtains results in less than 90 minutes, there is no need for the evaluation of normal mucosa biopsy for paired samples which leads to a decrease of time and cost for DNA preparation, and the analysis is objective and does not require highly trained personnel. Therefore, MicroSight® MSI can be easily implemented in every laboratory, even in an institute without a background of MSI evaluation.

## Supporting information

**S1 Table. PlentiPlex™ MSI PentaBase Panel primers.**
(DOCX)

**S2 Table. Software settings for MSI analysis on MyGo Pro using paired samples (and universal reference).**
(DOCX)

**S3 Table. Software settings for MSI analysis on BaseTyper™ using paired samples (and universal reference).**
(DOCX)

**S4 Table. Results from development cohort using paired samples and universal reference.**
(DOCX)

**S5 Table. Sensitivity and specificity using paired samples and universal reference for three extraction methods and two real-time PCR instruments.**
(DOCX)

**S6 Table. Sensitivity and specificity of paired samples and universal reference for five laboratories analyzing the same samples.**
(DOCX)

**S7 Table. 3x3 table from laboratory 2 using paired samples/universal reference.**
(DOCX)

**S8 Table. 3x3 table from laboratory 3 using paired samples/universal reference.**
(DOCX)

**S9 Table. 3x3 table from laboratory 4 using paired samples.**
(DOCX)

**S10 Table. 3x3 table from laboratory 4 using universal reference.**
(DOCX)

**S11 Table. 3x3 table from laboratory 5 using paired samples/universal reference.**
(DOCX)

**S12 Table. Deviating cases of paired samples from the validation cohort.**
(DOCX)

**S13 Table.**
(XLSX)

## Acknowledgments

We thank Alessia Paganotti and Alessandra di Pietro (Unit of Pathology, Department of Health Sciences, University of Eastern Piedmont, Novara, Italy), Carla Debernardi (Candiolo Cancer Institute, Fondazione del Piemonte per l'Oncologia, Candiolo, Italy), Renata Mariella Farioli (Molecular Pathology, Hospital of Legnano, Legnano (MI), Italy), and Stefano Ascani and Matteo Corsi (Laboratory of Molecular Oncology and Predictive Medicine, Pathology Unit, Azienda Ospedaliera S. Maria di Terni, Terni, Italy) for their techinal assistance during the inter-laboratory variation study as well as for their help with validating the MicroSight® MSI assay.

## Author Contributions

**Conceptualization:** Ulf Bech Christensen, Rasmus Koefoed Petersen.

**Data curation:** Kamilla Kolding Bendixen, Emeli Elisabeth Hansen, Sarah Kronborg Eriksen.

**Formal analysis:** Kamilla Kolding Bendixen, Michael Børgesen, Rasmus Koefoed Petersen.

**Funding acquisition:** Kamilla Kolding Bendixen, Ulf Bech Christensen, Rasmus Koefoed Petersen.

**Investigation:** Kamilla Kolding Bendixen, Giulia Dazio, Samantha Epistolio, Elisabetta Merlo, Renzo Boldorini, Tiziana Venesio, Alessandra Movilia, Cecilia Caprera.

**Methodology:** Kamilla Kolding Bendixen, Emeli Elisabeth Hansen, Sarah Kronborg Eriksen, Ulf Bech Christensen, Rasmus Koefoed Petersen.

**Project administration:** Kamilla Kolding Bendixen, Milo Frattini, Rasmus Koefoed Petersen.

**Resources:** Kamilla Kolding Bendixen, Giulia Dazio, Sarah Kronborg Eriksen, Samantha Epistolio, Elisabetta Merlo, Renzo Boldorini, Tiziana Venesio, Alessandra Movilia, Cecilia Caprera, Milo Frattini.

**Supervision:** Eva Christensen Arnspang, Milo Frattini, Rasmus Koefoed Petersen.

**Validation:** Kamilla Kolding Bendixen, Giulia Dazio, Samantha Epistolio, Elisabetta Merlo, Renzo Boldorini, Tiziana Venesio, Alessandra Movilia, Cecilia Caprera.

**Visualization:** Kamilla Kolding Bendixen, Milo Frattini, Rasmus Koefoed Petersen.

**Writing – original draft:** Kamilla Kolding Bendixen, Sofie Forsberg-Pho, Rasmus Koefoed Petersen.

**Writing – review & editing:** Kamilla Kolding Bendixen, Sofie Forsberg-Pho, Giulia Dazio, Samantha Epistolio, Milo Frattini, Rasmus Koefoed Petersen.

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
