## [Decision Letter · Decision Letter 0]

31 Jan 2024

PONE-D-23-37922One-instrument, objective microsatellite instability analysis using high-resolution meltPLOS ONE

Dear Dr. Petersen,

Thank you for submitting your manuscript to PLOS ONE. After careful consideration, we feel that it has merit but does not fully meet PLOS ONE’s publication criteria as it currently stands. Therefore, we invite you to submit a revised version of the manuscript that addresses the points raised during the review process.

**This manuscript was carefully reviewed by 2 experts. Both of them highly evaluated this study, but there are several minor issues which need to be addressed before acceptance. Please respond to each of the reviewer comments.**

We look forward to receiving your revised manuscript.

Kind regards,

Hiromu Suzuki, M.D., Ph.D.

Academic Editor

PLOS ONE

Journal Requirements:

 Please ensure that your manuscript meets PLOS ONE's style requirements, including those for file naming. The PLOS ONE style templates can be found at https://journals.plos.org/plosone/s/file?id=wjVg/PLOSOne_formatting_sample_main_body.pdf and https://journals.plos.org/plosone/s/file?id=ba62/PLOSOne_formatting_sample_title_authors_affiliations.pdf 2. Thank you for stating in your Funding Statement: This work was supported by Innovation Fund Denmark (www.innovationsfonden.dk) [grant number 8062-00374B and 9078-00239B] in the form of research materials and salaries (K.K.B., E.E.H., S.K.E., M.B., U.B.C., and R.K.P). Innovation Fund Denmark did not have any additional role in the study design, data collection and analysis, decision to publish, or preparation of the manuscript.  Please provide an amended statement that declares all the funding or sources of support (whether external or internal to your organization) received during this study, as detailed online in our guide for authors at http://journals.plos.org/plosone/s/submit-now.  Please also include the statement “There was no additional external funding received for this study.” in your updated Funding Statement. Please include your amended Funding Statement within your cover letter. We will change the online submission form on your behalf. 3. Thank you for stating the following in the Competing Interests section: I have read the journal's policy and the authors of this manuscript have the following competing interests: K.K.B., E.E.H., S.K.E., M.B., U.B.C., and R.K.P. received salaries from Innovation Fund Denmark. The funding organization did not play a role in the study design, data collection and analysis, decision to publish, or preparation of the manuscript and only provided financial support in the form of authors’ salaries and research materials. K.K.B., S.FP., E.E.H., S.K.E., M.B., U.B.C., and R.K.P. were employed at PentaBase A/S during this study. PentaBase A/S is the manufacturer and seller of MicroSight® MSI 1-step HRM Analysis, which is the assay of interest in this study. U.B.C. and R.K.P. are co-owners of PentaBase A/S. K.K.B., E.E.H., S.K.E., U.B.C., and R.K.P. are all founders of a patent application covering the novel technology which MicroSight® MSI 1-step HRM Analysis is based on [International Publication number: WO2020229510A1]. The remaining authors declare no conflict of interest. This does not alter our adherence to PLOS ONE policies on sharing data and materials.    We note that one or more of the authors are employed by a commercial company. a. Please provide an amended Funding Statement declaring this commercial affiliation, as well as a statement regarding the Role of Funders in your study. If the funding organization did not play a role in the study design, data collection and analysis, decision to publish, or preparation of the manuscript and only provided financial support in the form of authors' salaries and/or research materials, please review your statements relating to the author contributions, and ensure you have specifically and accurately indicated the role(s) that these authors had in your study. You can update author roles in the Author Contributions section of the online submission form. Please also include the following statement within your amended Funding Statement. “The funder provided support in the form of salaries for authors, but did not have any additional role in the study design, data collection and analysis, decision to publish, or preparation of the manuscript. The specific roles of these authors are articulated in the ‘author contributions’ section.”If your commercial affiliation did play a role in your study, please state and explain this role within your updated Funding Statement.  b. Please also provide an updated Competing Interests Statement declaring this commercial affiliation along with any other relevant declarations relating to employment, consultancy, patents, products in development, or marketed products, etc.   Within your Competing Interests Statement, please confirm that this commercial affiliation does not alter your adherence to all PLOS ONE policies on sharing data and materials by including the following statement: "This does not alter our adherence to  PLOS ONE policies on sharing data and materials.” (as detailed online in our guide for authors http://journals.plos.org/plosone/s/competing-interests) . If this adherence statement is not accurate and  there are restrictions on sharing of data and/or materials, please state these. Please note that we cannot proceed with consideration of your article until this information has been declared. Please include both an updated Funding Statement and Competing Interests Statement in your cover letter. We will change the online submission form on your behalf. 4. We note that Figures 2 and 3 in your submission contain copyrighted images. All PLOS content is published under the Creative Commons Attribution License (CC BY 4.0), which means that the manuscript, images, and Supporting Information files will be freely available online, and any third party is permitted to access, download, copy, distribute, and use these materials in any way, even commercially, with proper attribution. For more information, see our copyright guidelines: http://journals.plos.org/plosone/s/licenses-and-copyright. We require you to either present written permission from the copyright holder to publish these figures specifically under the CC BY 4.0 license, or remove the figures from your submission: a. You may seek permission from the original copyright holder of Figures 2 and 3 to publish the content specifically under the CC BY 4.0 license. We recommend that you contact the original copyright holder with the Content Permission Form (http://journals.plos.org/plosone/s/file?id=7c09/content-permission-form.pdf) and the following text:“I request permission for the open-access journal PLOS ONE to publish XXX under the Creative Commons Attribution License (CCAL) CC BY 4.0 (http://creativecommons.org/licenses/by/4.0/). Please be aware that this license allows unrestricted use and distribution, even commercially, by third parties. Please reply and provide explicit written permission to publish XXX under a CC BY license and complete the attached form.”Please upload the completed Content Permission Form or other proof of granted permissions as an "Other" file with your submission.  In the figure caption of the copyrighted figure, please include the following text: “Reprinted from [ref] under a CC BY license, with permission from [name of publisher], original copyright [original copyright year].”b. If you are unable to obtain permission from the original copyright holder to publish these figures under the CC BY 4.0 license or if the copyright holder’s requirements are incompatible with the CC BY 4.0 license, please either i) remove the figure or ii) supply a replacement figure that complies with the CC BY 4.0 license. Please check copyright information on all replacement figures and update the figure caption with source information. If applicable, please specify in the figure caption text when a figure is similar but not identical to the original image and is therefore for illustrative purposes only. 5. PLOS requires an ORCID iD for the corresponding author in Editorial Manager on papers submitted after December 6th, 2016. Please ensure that you have an ORCID iD and that it is validated in Editorial Manager. To do this, go to ‘Update my Information’ (in the upper left-hand corner of the main menu), and click on the Fetch/Validate link next to the ORCID field. This will take you to the ORCID site and allow you to create a new iD or authenticate a pre-existing iD in Editorial Manager. Please see the following video for instructions on linking an ORCID iD to your Editorial Manager account: https://www.youtube.com/watch?v=_xcclfuvtxQ 6. 
Please review your reference list to ensure that it is complete and correct. If you have cited papers that have been retracted, please include the rationale for doing so in the manuscript text, or remove these references and replace them with relevant current references. Any changes to the reference list should be mentioned in the rebuttal letter that accompanies your revised manuscript. If you need to cite a retracted article, indicate the article’s retracted status in the References list and also include a citation and full reference for the retraction notice.

Additional Editor Comments:

This manuscript was carefully reviewed by 2 experts. Both of them highly evaluated this study, but there are several minor issues which need to be addressed before acceptance. Please respond to each of the reviewer comments.

Reviewers' comments:

Reviewer's Responses to Questions

**Comments to the Author**

1. Is the manuscript technically sound, and do the data support the conclusions?

Reviewer #1: Yes

Reviewer #2: Yes

2. Has the statistical analysis been performed appropriately and rigorously? 

Reviewer #1: Yes

Reviewer #2: Yes

3. Have the authors made all data underlying the findings in their manuscript fully available?

Reviewer #1: Yes

Reviewer #2: Yes

4. Is the manuscript presented in an intelligible fashion and written in standard English?

Reviewer #1: Yes

Reviewer #2: Yes

5. Review Comments to the Author

Reviewer #1: 1. I thought the author's point in this paper was that MicroSight® MSI 1-Step HRM Analysis, which can evaluate MSI without fragment analysis, could be as good as MSI testing using traditional fragment analysis. This point is not clear in this paper, and I would recommend that the paper be revised. Similarly, the authors should present the results of the investigation of whether there is a difference in the instability of microsatellite markers (BAT-25, BAT-26, NR-22, NR-24, MONO27) between conventional fragment analysis methods and MicroSight® MSI 1-Step HRM Analysis. It would be trivial to compare MicroSight® MSI 1-Step HRM Analysis with other MSI testing or to investigate whether differences in DNA extraction methods affect the MicroSight® MSI 1-Step HRM Analysis.

2. Many reports have evaluated microsatellite instability using BAT-25, BAT-26, NR-21, NR-24, and NR-27 (e.g., Ann Oncol. 2019 Aug 1;30(8):1232-1243. doi: 10.1093). Please describe why the authors used NR-22 instead of NR-21.

Reviewer #2: The manuscript entitled "One-instrument, objective microsatellite instability analysis using high-resolution melt" highlighted that the MicroSight® MSI assay showed a high repeatability regardless of DNA extraction method and PCR platform, and a 100% agreement of the MSI status with the PCR fragment analysis method.

- The AUthors should provide the expand forms for all acronyms, including gene acronyms, through the text when they first appear.

- Gene acronyms should be written in italics.

- The Authors should consider to cite the role of pre-analitics in the discussion (PMID: 34440647, PMID: 32887373)

6. PLOS authors have the option to publish the peer review history of their article (what does this mean?). If published, this will include your full peer review and any attached files.

Reviewer #1: No

Reviewer #2: No

---

## [Author Response · Author response to Decision Letter 0]

6 Mar 2024

1. The “Abstract” heading was changed from 20pt font to 18pt font.

2. The “1. Introduction” heading was changed from 20pt font to 18pt font.

3. References were cited in brackets [] instead of parentheses ()

4. The names of the figure files were changed from “Fig-1.tif”, “Fig-2.tif”, “Fig-3.tif”, “Fig-4.tif”, and “Fig-5.tif” to “Fig1.tif”, “Fig2.tif”, “Fig3.tif”, “Fig4.tif”, and “Fig5.tif”.

5. The supporting information was split from 1 to 12 documents, and the name of the supporting information file was changed from “Supporting information.docx” to “S1_File.docx”, “S2_File.docx”, “S3_File.docx”, “S4_File.docx”, “S5_File.docx”, “S6_File.docx”, “S7_File.docx”, “S8_File.docx”, “S9_File.docx”, “S10_File.docx”, “S11_File.docx”, and “S12_File.docx”. 

1. The name of the paper was centralized on the first page. 

2. Corresponding author was changed from Kamilla Kolding Bendixen and Sofie Forsberg-Pho to Rasmus Koefoed Petersen.

3. Milo Frattini and Rasmus Koefoed Petersen were changed from 2nd set of equal contributors to 1st set of equal contributors.

4. “*Rasmus Koefoed Petersen”, “E-mail:rkp@pentabase.com (RKP)”, and “¶These authors contributed equally to this work” were added below the affiliation section

5. E-mails from all authors were removed.

This work was supported by Innovation Fund Denmark (www.innovationsfonden.dk) [grant number 8062-00374B and 9078-00239B] in the form of research materials and salaries (K.K.B., E.E.H., S.K.E., M.B., U.B.C., and R.K.P). Innovation Fund Denmark did not have any additional role in the study design, data collection and analysis, decision to publish, or preparation of the manuscript. 

Our Funding Statement has been changed to the following: “This work was supported by Innovation Fund Denmark (https://innovationsfonden.dk/en) [grant number 8062-00374B received and 9078-00239B both received by U.B.C.] in the form of research materials and salaries (K.K.B., E.E.H., S.K.E., M.B., U.B.C., and R.K.P). The funder had no role in study design, data collection and analysis, decision to publish, or preparation of the manuscript. K.K.B., S.FP., E.E.H., S.K.E., M.B., U.B.C., and R.K.P. were employed at PentaBase A/S during this study and received salary from the company. PentaBase A/S is the manufacturer and seller of MicroSight® MSI 1-step HRM Analysis, which is the assay of interest in this study. U.B.C. and R.K.P. are shareholders and board members of PentaBase A/S. There was no additional external funding received for this study.” This has furthermore been included in our cover letter.

I have read the journal's policy and the authors of this manuscript have the following competing interests: K.K.B., E.E.H., S.K.E., M.B., U.B.C., and R.K.P. received salaries from Innovation Fund Denmark. The funding organization did not play a role in the study design, data collection and analysis, decision to publish, or preparation of the manuscript and only provided financial support in the form of authors’ salaries and research materials. K.K.B., S.FP., E.E.H., S.K.E., M.B., U.B.C., and R.K.P. were employed at PentaBase A/S during this study. PentaBase A/S is the manufacturer and seller of MicroSight® MSI 1-step HRM Analysis, which is the assay of interest in this study. U.B.C. and R.K.P. are co-owners of PentaBase A/S. K.K.B., E.E.H., S.K.E., U.B.C., and R.K.P. are all founders of a patent application covering the novel technology which MicroSight® MSI 1-step HRM Analysis is based on [International Publication number: WO2020229510A1]. The remaining authors declare no conflict of interest. This does not alter our adherence to PLOS ONE policies on sharing data and materials.

We note that one or more of the authors are employed by a commercial company.

“The funder provided support in the form of salaries for authors, but did not have any additional role in the study design, data collection and analysis, decision to publish, or preparation of the manuscript. The specific roles of these authors are articulated in the ‘author contributions’ section.”

This has been done, see the comment reported for point 2 “Thank you for stating in your Funding Statement”.

The Competing Interests Statement has been changed to the following: “K.K.B., E.E.H., S.K.E., M.B., U.B.C., and R.K.P. received salaries from Innovation Fund Denmark. The funder had no role in study design, data collection and analysis, decision to publish, or preparation of the manuscript and only provided financial support in the form of authors’ salaries and research materials. K.K.B., S.FP., E.E.H., S.K.E., M.B., U.B.C., and R.K.P. were employed at PentaBase A/S during this study. PentaBase A/S is the manufacturer and seller of MicroSight® MSI 1-step HRM Analysis, which is the assay of interest in this study. U.B.C. and R.K.P. are shareholders and board members of PentaBase A/S. K.K.B., E.E.H., S.K.E., U.B.C., and R.K.P. are all founders of a patent application covering the novel technology which MicroSight® MSI 1-step HRM Analysis is based on [International Publication number: WO2020229510A1 (Melting temperature methods, kits and reporter oligo for detecting variant nucleic acids)]. The remaining authors declare no conflict of interest. This does not alter our adherence to PLOS ONE policies on sharing data and materials.” This has furthermore been included in our cover letter.

4. We note that Figures 2 and 3 in your submission contain copyrighted images. All PLOS content is published under the Creative Commons Attribution License (CC BY 4.0), which means that the manuscript, images, and Supporting Information files will be freely available online, and any third party is permitted to access, download, copy, distribute, and use these materials in any way, even commercially, with proper attribution. For more information, see our copyright guidelines: http://journals.plos.org/plosone/s/licenses-and-copyright.

We require you to either present written permission from the copyright holder to publish these figures specifically under the CC BY 4.0 license, or remove the figures from your submission:

a. You may seek permission from the original copyright holder of Figures 2 and 3 to publish the content specifically under the CC BY 4.0 license. 

All figures were generated in R studio by Kamilla Kolding Bendixen, and none of them have any copyright on them.

The corresponding author has created an ORCID iD: 0009-0003-6889-4899.

We have checked the reference list, thank you.

Reviewer 1:

1. I thought the author's point in this paper was that MicroSight® MSI 1-Step HRM Analysis, which can evaluate MSI without fragment analysis, could be as good as MSI testing using traditional fragment analysis. This point is not clear in this paper, and I would recommend that the paper be revised. Similarly, the authors should present the results of the investigation of whether there is a difference in the instability of microsatellite markers (BAT-25, BAT-26, NR-22, NR-24, MONO27) between conventional fragment analysis methods and MicroSight® MSI 1-Step HRM Analysis. It would be trivial to compare MicroSight® MSI 1-Step HRM Analysis with other MSI testing or to investigate whether differences in DNA extraction methods affect the MicroSight® MSI 1-Step HRM Analysis.

Response: We agree with the reviewer that a major point of the manuscript is to compare the suggested method to the current fragment analysis gold standard. The text has been slightly changed in the last section of the introduction and in the conclusion to highlight that the clinical validation is indeed one-to-one comparison with traditional fragment analysis: please refer to lines 52-53 in the introduction and lines 482-483 in the conclusions.

2. Many reports have evaluated microsatellite instability using BAT-25, BAT-26, NR-21, NR-24, and NR-27 (e.g., Ann Oncol. 2019 Aug 1;30(8):1232-1243. doi: 10.1093). Please describe why the authors used NR-22 instead of NR-21.

Response: NR-22 has as part of a 5 loci panel (with BAT-25, BAT-26, NR-21 and NR-24) been suggested for evaluation of MSI status. The reason for choosing this marker was 1) sensitivity (some studies indicates that NR-22 are among the most sensitive micro-satellite markers (eg Xicola et al 2007, https://doi.org/10.1093/jnci/djk033), well aware that other studies do not completely agree, 2) the relative monomorphic nature of the loci (Suraweera et al. 2002, doi 10.1053/gast.2002.37070 ), enabling the application of a universal reference and 3) the technical and clinical performance as illustrated by the unique separation of MSS and MSI cases presented also in the current manuscript (for example refer to Fig 3). This concept was reported in the discussion section, please see lines 411-414. 

Reviewer 2:

- The Authors should provide the expand forms for all acronyms, including gene acronyms, through the text when they first appear.

Response: Expanded forms for protein names and institutions have been introduced to the introduction section. Although not used in latter part of the manuscript abbreviation is kept as some of the proteins and institutions will be normally recognized by the abbreviation. For the naming of microsatellite loci, we have used only the short names as we believe they will be better recognized hereby.

- Gene acronyms should be written in italics.

Response: We thank the reviewer for the comment, and we have revised the text accordingly.

- The Authors should consider to cite the role of pre-analitics in the discussion (PMID: 34440647, PMID: 32887373)

Response: We are completely in line with the reviewer’s comment. Therefor part of the study also includes evaluation of different purification methods to investigate the methods ability to handle various DNA qualities and eluents. Further, a paragraph mentioning the importance of DNA sample quality when evaluating MSI status has been added to the introduction and supported by the addition of the reference Malapelle et al 2020. Accordingly, we have modified the text in the following lines: 85-87, 443-444

---

## [Decision Letter · Decision Letter 1]

1 Apr 2024

One-instrument, objective microsatellite instability analysis using high-resolution melt

PONE-D-23-37922R1

Dear Dr. Petersen,

We’re pleased to inform you that your manuscript has been judged scientifically suitable for publication and will be formally accepted for publication once it meets all outstanding technical requirements.

Kind regards,

Hiromu Suzuki, M.D., Ph.D.

Academic Editor

PLOS ONE

Additional Editor Comments (optional):

The authors addressed all issues raised by the reviewer.

Reviewers' comments:

Reviewer's Responses to Questions

**Comments to the Author**

1. If the authors have adequately addressed your comments raised in a previous round of review and you feel that this manuscript is now acceptable for publication, you may indicate that here to bypass the “Comments to the Author” section, enter your conflict of interest statement in the “Confidential to Editor” section, and submit your "Accept" recommendation.

Reviewer #1: All comments have been addressed

Reviewer #2: All comments have been addressed

2. Is the manuscript technically sound, and do the data support the conclusions?

Reviewer #1: Yes

Reviewer #2: Yes

3. Has the statistical analysis been performed appropriately and rigorously? 

Reviewer #1: Yes

Reviewer #2: Yes

4. Have the authors made all data underlying the findings in their manuscript fully available?

Reviewer #1: Yes

Reviewer #2: Yes

5. Is the manuscript presented in an intelligible fashion and written in standard English?

Reviewer #1: Yes

Reviewer #2: Yes

6. Review Comments to the Author

Reviewer #1: (No Response)

Reviewer #2: The authors have addressed all my concerns and I have no further comments.

The authors have addressed all my concerns and I have no further comments.

The authors have addressed all my concerns and I have no further comments.

7. PLOS authors have the option to publish the peer review history of their article (what does this mean?). If published, this will include your full peer review and any attached files.

Reviewer #1: No

Reviewer #2: No
